# Mitochondrial Redox Metabolism: The Epicenter of Metabolism during Cancer Progression

**DOI:** 10.3390/antiox10111838

**Published:** 2021-11-19

**Authors:** Feroza K. Choudhury

**Affiliations:** Drug Metabolism and Pharmacokinetics Department, Genentech Inc., South San Francisco, CA 94080, USA; choudhury.feroza@gene.com

**Keywords:** mitochondrial redox metabolism, ROS signaling, tumor development, metastasis, distant colonization

## Abstract

Mitochondrial redox metabolism is the central component in the cellular metabolic landscape, where anabolic and catabolic pathways are reprogrammed to maintain optimum redox homeostasis. During different stages of cancer, the mitochondrial redox status plays an active role in navigating cancer cells’ progression and regulating metabolic adaptation according to the constraints of each stage. Mitochondrial reactive oxygen species (ROS) accumulation induces malignant transformation. Once vigorous cell proliferation renders the core of the solid tumor hypoxic, the mitochondrial electron transport chain mediates ROS signaling for bringing about cellular adaptation to hypoxia. Highly aggressive cells are selected in this process, which are capable of progressing through the enhanced oxidative stress encountered during different stages of metastasis for distant colonization. Mitochondrial oxidative metabolism is suppressed to lower ROS generation, and the overall cellular metabolism is reprogrammed to maintain the optimum NADPH level in the mitochondria required for redox homeostasis. After reaching the distant organ, the intrinsic metabolic limitations of that organ dictate the success of colonization and flexibility of the mitochondrial metabolism of cancer cells plays a pivotal role in their adaptation to the new environment.

## 1. Introduction

The dynamics of oxidative stress navigates the cancer cell throughout its progression in different stages- where higher reactive oxygen species (ROS) favor malignant transformation, then decrease during tumor formation to the optimum level required for maintaining cellular redox homeostasis. It again increases during hypoxia and selects for aggressive metastatic cells. These cells are capable of withstanding the highly stressful steps of metastasis and the oxidative environment of the blood circulation. The redox status of the metastatic cells dictates successful colonization in distant organs [1,2,3]. Technical challenges in their detection and modulation in vivo have greatly limited our understanding on their role [4]. The dynamics of ROS during cancer progression are reciprocated in the level of TIGAR (TP53-induced glycolysis and apoptosis regulator), which is the P53-induced modulator for cellular adaptation to oxidative stress-increasing NADPH production and heightening the antioxidant defense. In pancreatic ductal adenocarcinoma development, during the early stages, high TIGAR levels are needed to mitigate the ROS associated with transformation. Then, as the tumor progresses, decreasing levels of TIGAR appear to increase the malignancy of cancer cells consistent with the selection for invasive cells with a higher ROS scavenging capacity. At a later stage, the TIGAR levels go up again to buffer the oxidative stress experienced by metastatic cells [5]. Cancer cells need to develop higher ROS scavenging capabilities to maintain their required level for initiation, adaptation and progression and mitigate toxic accumulation [6,7,8].

The alteration of oxidative stress is associated with the reprogramming of cellular metabolism during different stages of cancer progression to maintain redox homeostasis and support the metabolic demand imposed in each stage [9]. Mitochondrial redox metabolism resides at the epicenter of this process and orchestrates different aspects of cellular metabolism for adaptation and, in association with generating ROS signaling actively, drives cancer cells’ progression through tumor development, metastasis and colonization in distant organs [10,11]. The role of mitochondria in cancer development is primarily focused on mutations in genes in TCA cycle enzymes, namely succinate dehydrogenase (SDH), fumarate hydratase (FH) and isocitrate dehydrogenase 2 (IDH2), leading to the accumulation of succinate, fumarate and 2-hydroxyglycerate (2HG). These metabolites mediate epigenetic changes leading to the development of some subtypes of cancer. When the mitochondrion is viewed in the context of its role in ROS signaling and reprogramming cellular metabolism, its impact is much more complex and substantial in different stages across all types of cancer [12,13,14]. This review focuses on the various aspects of mitochondrial redox metabolism and ROS generation during different stages of tumor development and metastasis, with an emphasis on how it plays a central role in reprogramming cellular metabolism for efficient transition and adaptation in each stage.

## 2. Mitochondrial Redox Metabolism

The mitochondrial electron transport chain (ETC)-mediated inner membrane electrochemical gradient drives oxidative phosphorylation for energy production. It is also the primary component for generating mitochondrial reactive oxygen species (ROS) and maintaining the redox balance (Figure 1). In a situation where the sequential movement of electrons through the ETC is delayed or halted or ATP synthase is inhibited, creating increased membrane hyperpolarization, it causes the electrons at complexes I, II and III to interact with O_2_ to form a superoxide [2,15]. Due to the structural organization, complex III is the primary site that can produce superoxide in the intermembrane space. The Fenton reaction produces a highly reactive hydroxyl radical via the reaction between Fe(II) and hydrogen per oxide (H_2_O_2_). This may interact with lipid species and generate lipid peroxyl radicals. Several enzymes in the mitochondria that participate in oxidation-reduction reactions, namely the 2-oxoglutarate dehydrogenase (OGDH), branched-chain 2-oxoacid dehydrogenase (BCKDH) and pyruvate dehydrogenase (PDH) complexes, may also contribute to ROS generation [16].

ROS can also be produced in a regulated manner for signaling via NADPH oxidases (NOX proteins), which generates a superoxide from O_2_ while oxidizing NADPH. Among seven catalytic isoforms, NOX1, NOX2, NOX4 and NOX5 are associated with cancer development, having different subcellular locations. NOX-derived ROS signaling induces proliferation upon growth factor stimulation and promotes angiogenesis, cell migration and invasion [17]. NOX4 localized in the inner mitochondrial membrane can produce ROS, which can directly oxidize mitochondrial complex subunits, as well as mediate mitochondrial DNA damage [18]. It also acts as an ATP sensor and mediates metabolic reprogramming via regulating pyruvate kinase-M2 (PKM2) stability [19].

Mitochondrial superoxide is dismutated to H_2_O_2_ by manganese superoxide dismutase (MnSOD/SOD2) in the matrix and by Cu/Zn superoxide dismutase (Cu/ZnSOD/SOD1) in the intermembrane space and cytosol. H_2_O_2_ acts as diffusible signaling molecule and is neutralized for water by enzymes, including peroxiredoxin-thioredoxin and glutathione peroxidases (GPX) [20]. Recent studies have drawn attention towards the pathological significance of mitochondrial peroxiredoxins, as their expression is associated with the initiation and progression of multiple types of cancers [21]. These antioxidant systems depend on NADPH as the final electron donor, and hence, the redox balance of mitochondria relies on their NADP^+^/NADPH ratio.

NADPH is primarily generated in mitochondria via mitochondrial inner membrane enzyme nicotinamide nucleotide transhydrogenase (NNT), which catalyzes the transfer of a hydride from NADH to NADP^+^, producing NADPH and NAD^+^, coupled to the translocation of one proton across the membrane (Figure 1) [22]. Since this enzyme is energized by a mitochondrial inner membrane electrochemical gradient and uses NADH as a reducing equivalent, it ensures the mitochondrial metabolism is dedicated to maintaining the optimum NADPH level in mitochondria required for antioxidant defense, as well as the biosynthesis of macromolecules. In a recent study, where they employed a genetically encoded tool to perturb the NAD^+^/NADH and NADP^+^/NADPH ratios in the mitochondria and cytosol, they found that the redox states of mitochondrial NADPH and NADH pools are connected, but they are asymmetrical. The oxidation of NADP^+^/NADPH in the mitochondria leads to the oxidation of NAD^+^/NADH, but the reverse is not true. This, along with the observation that perturbing the mitochondrial NADP^+^/NADPH ratio increases the TCA cycle activity for the generation of NADH, indicates that the mitochondrial metabolism tends towards maintaining the optimum NADP^+^/NAPDH ratio [23]. Noteworthy to mention is that a similar connection has not been found for the cytosolic NADPH and NADH pools. Analyzing compartmentalized NADPH production revealed that NADH supports NADPH production in the mitochondria but not in the cytosol [24]. Since NNT plays a direct role in consuming NADH to generate NADPH and NAD^+^, it is the central player in integrating mitochondrial energy metabolism to maintain the redox balance. For this, NNT is considered a sensor of mitochondrial biology [25,26].

Other enzymes that play an important role in generating mitochondrial NADPH are mitochondrial TCA cycle enzyme isocitrate dehydrogenase 2 (IDH2), catalyzing the oxidative decarboxylation of isocitrate to α-ketoglutarate, and mitochondrial malic enzymes 2 and 3 (ME2 and ME3), catalyzing the oxidative decarboxylation of malate to pyruvate (Figure 1) [27,28]. Serine catabolism via the mitochondrial one-carbon metabolism pathway is also important to generate NADPH (Figure 1) [29]. Mitochondrial NAD kinase (NADK2) phosphorylates NAD^+^ to NADP^+^, which is required for NADPH production. The loss of NADK2 is associated with reduced respiration and increased intracellular ROS, but its role in maintaining the redox balance and cancer is yet to be investigated [30,31]. One important concept that has emerged is that the maintenance of mitochondrial redox metabolism is not a separate event; rather, it is intricately tied with energy metabolism and biosynthetic processes and navigates metabolic reprogramming, bringing change to the mitochondrial biology and overall cell fate.

Sirtuin proteins, specifically SIRT3 residing in mitochondria, act as a key regulator of mitochondrial redox metabolism. It functions as a NAD^+^-dependent protein deacetylase, which gets activated in high NAD^+^/NADH conditions. Thus, it acts as a sensor of the mitochondrial metabolic state and activates pathways for maintaining the optimum ATP and NADPH levels [32,33,34]. It reduces ROS generation from ETC and enhances the detoxification through the activation of antioxidant enzymes, like SOD2, resulting in maintenance of the redox balance in mitochondria [35,36,37].

A critical role of mitochondrial redox metabolism is to support proline biosynthesis as it consumes NAD(P)H [31]. NAD^+^ generation during proline biosynthesis uncouples the TCA cycle from respiration and could help minimize the generation of ROS from ETC [38]. The proline shuttle mediated by proline oxidase (POX) is important for transferring redox equivalents between the mitochondria and cytosol and generating ROS for signaling [39,40].

How far the cytosolic pathways support maintaining the redox balance in mitochondria is an area of active research. A compartmentalized NAD^+^/NADH ratio is maintained in the mitochondria (NAD^+^/NADH ratio 7 to 8) and cytosol (NAD^+^/NADH ratio 60–700) mainly by malate–aspartate shuttle, malate–citrate shuttle and glycerophosphate shuttle [41,42]. Cytosolic reductive carboxylation comes into play for supporting mitochondrial NADPH generation for mitigating mitochondrial ROS. Here, IDH1 consumes NADPH from cytosol-generating citrate, which goes inside the mitochondria for oxidation, where IDH2 generates NADPH [43]. This cycle can also work in reverse and follow the path of mitochondrial reductive carboxylation to support cytosolic NADPH production [44]. In mitochondrial oxidative stress, the directionality of the cycle is maintained towards the generation of NADPH in the mitochondria through regulation of the IDH2 enzyme [45].

In addition, mitochondrial plasticity comes into action when cells experience stress. Fusion helps to alleviate stress by complementing the content of two damaged mitochondria and, thus, maximize the oxidative capacity. Fission is needed to generate new mitochondria but, during stress, helps to remove damaged mitochondria as a measure of the quality control [46]. Mitochondrial fission is also responsible for the generation of ROS in a hyperglycemic condition, and the induction of fusion can reduce ROS accumulation [47].

## 3. Malignant Transformation

Mitochondrial DNA mutations reported in a wide range of cancers and associated with impaired oxidative phosphorylation (OXPHOS) stimulate malignant transformation via generating ROS [48,49]. The first evidence of mitochondrial ROS-induced tumorigenicity was observed from a swap experiment where mtDNA of a prostate cancer cell was exchanged with pathogenic mtDNA [50]. Pathogenic mtDNA mutations arise in tumors almost at a similar rate to mutations in most common cancer driver genes, with complex I accumulating the highest loss of function mutations [51]. Apart from mutations impairing OXPHOS, mutations in other mitochondrial enzymes may also be responsible for the accumulation of mitochondrial ROS. One such case is the cancer cells deficient in fumarate hydratase (FH), where the accumulation of fumarate results in conjugation with glutathione to produce succinate glutathione. This acts as an alternate substrate to glutathione reductase, exhausting NADPH in the process and causing the accumulation of mitochondrial ROS [52]. Mitochondrial ROS generation can induce tumor formation in ways similar to growth factor activation, like the phosphoinositide 3-kinase (PI3K) pathway, central to the growth factor response. Mitochondrial ROS inactivates its antagonist tumor suppressor PTEN (phosphatase and tensin homolog deleted from chromosome 10), promoting the PI3K-signaling cascade [53,54].

Mitochondrial ROS signaling is a critical aspect of some oncogenic drivers for the initiation of malignant transformation and control metabolic adaptation during various stages of tumorigenesis. Notable among them is Kras dependent on mitochondrial ROS generation for cancer initiation and progression corresponding with a metabolic switch from oxidative phosphorylation to glycolysis [55,56]. Ras-induced transformation is also associated with alterations of the mitochondrial dynamics promoting fission [57].

Failure of the mitochondrial ROS scavenging system may also promote tumorigenesis. Mitochondrial ROS is responsible for signaling cell proliferation and quiescence where MnSOD serves as a switch. Decreased MnSOD activity favors proliferation by increasing the accumulation of superoxide, and increasing the activity of MnSOD facilitates the proliferating cell transition into quiescence [58,59]. The accumulation of mitochondrial ROS caused by a lack of MnSOD activity leads to an increase of double-stranded breaks and chromosomal translocation, causing genomic instability and promoting oncogenic transformation [60,61].

During the early phage of tumorigenesis, ROS appears to be mutagenic and, therefore, supports transformation [62,63]. ROS increases upon transformation and is quickly scavenged with the concurrent activation of antioxidant programs, primarily mediated by Nrf2. The investigation of ROS metabolism upon expression of the endogenous oncogenic alleles of *Kras*, *Braf* and *Myc* showed that ROS are actively suppressed through the Nrf2-mediated antioxidant response [64]. The ROS-dependent inactivation of PKM2 (pyruvate kinase M2) diverts glucose carbon to the oxidative pentose phosphate pathway (PPP), resulting in increased NAPDH production. This is one of the distinct mechanisms of ROS signaling-induced metabolic reprogramming for maintaining the redox balance [65].

When mitochondrial ROS generation is too high, it can be toxic to the cell, and cellular apoptosis and necrosis comes into play, inducing cell death. One of the ways mtDNA mutation in ATP synthase subunit 6 gene (MTATP6) promotes tumor growth is by preventing apoptosis [66].

One aspect of mitochondrial metabolism that contributes to maintaining the redox homeostasis during cancer initiation is through its control of glutathione peroxidase [67]. Mitochondrial enzyme glutamate dehydrogenase (GDH1), upregulated in many cancers, is important for maintaining the intracellular level of its product, α-ketoglutarate, and its subsequent metabolite, fumarate, which binds to and activates the ROS-scavenging enzyme glutathione peroxidase 1. Targeting GDH1 results in the depletion of fumarate, thus inhibiting glutathione-mediated ROS scavenging, and it has shown therapeutic potential in vivo in lung cancer models [68].

Mitochondria harboring mutation in the enzymes of the TCA cycle or ETC resulting in the defective oxidative function of mitochondria relies on reductive carboxylation to utilize glutamine for lipid and macromolecule synthesis [69].

## 4. Tumor Development

During aggressive cell proliferation leading to a clinically detectable tumor, mitochondrial redox metabolism comes into play, regulating various aspects of tumor development, ranging from nutrient uptake to invasiveness. Cancer has been regarded as a metabolic disease arising from impaired mitochondrial oxidative metabolism for almost 100 years, from the initial observation of Otto Warburg about a lack of proportion between glycolysis and respiration, where cells convert glucose to lactate for excretion and commits to an inefficient way of ATP synthesis even when there is sufficient oxygen for respiration [70]. Furthermore, the observation of a “spare respiratory capacity” in cancer cells indicates that glucose carbon is diverted to lactate without reaching the maximum respirational potential [71]. The increased consumption of glucose associated with unrestrained cell proliferation, characteristic of most cancer cells, imposes demands on NAD^+^ for maintaining a higher glycolytic flux and subsequent metabolism of glucose [72]. A recent study suggested that an increased demand for NAD^+^ relative to mitochondrial ATP production is what drives cells to commit to aerobic glycolysis [73]. Another major fuel for cancer cell glutamine, regulated by oncogene *Myc*, is also excreted as lactate and alanine to support the redox balance via generating NADPH via the cytosolic malic enzyme [74,75,76]. Glutamine contributes to TCA cycle anaplerosis and regenerates NAD^+^ via the action of cytosolic malate dehydrogenase (MDH1), enabling glucose carbon to be diverted from energy production to anabolic processes [77,78].

Cancer cells’ commitment towards aerobic glycolysis vs. mitochondrial oxidation depends largely on the tissue from which the cancer originated rather than their oncogenic drivers. This preference for the bioenergetic route has a pronounced impact on the type of fuel consumption. Recent ^13^C-glucose infusion studies in patients showed that Warburg-like metabolic reprogramming is observed in clear cell renal cell carcinoma, making cells dependent on glycolysis for energy production [79]. Contrary to this, robust glucose oxidation is a major feature in tumors in the brain and non-small cell lung cancer (NSCLC) patients [80,81]. When investigating the mitochondrial metabolism in the context of tumor architecture, an even more complex picture arises, as tumors contain oxygenated and hypoxic regions based on tissue perfusion, which contributes to the already heterogeneous tumor metabolism [82]. Even though the poorly vascularized core of the solid tumor has limited oxygen availability, some cancers still rely on mitochondrial respiration for ATP generation, as ETC can function optimally at oxygen levels as low as 0.5% [83]. In NSCLC patients, the glucose metabolism is heterogeneous within and between tumors, depending on the tissue perfusion, which determines the contribution of various fuels to the TCA cycle, with less-perfused regions depending on glucose and well-perfused regions using alternate fuels [81]. Circulating lactate is the major contributor to TCA cycle intermediates in most tissues, except in cases like pancreatic cancer, which prefers glutamine over lactate to feed the TCA cycle [84]. Oncologic behavior is also associated with fuel consumption, with aggressive NSCLC tumors taking lactate as a respiratory fuel from plasma via monocarboxylate transporter 1 (MCT1) [85].

Oxygenated tumor cells use lactate as a prominent respiratory substrate, while hypoxic tumor cells primarily follow a Warburg-like metabolism, converting glucose to lactate [86]. Thus, the glycolytic and oxidative subpopulations of cancer cells in a tumor exchange lactate to support growth [87]. Tumors can also exploit the metabolism of associated stroma and consume fuel from them in a phenomenon termed as “Stromal–Epithelial Metabolic Coupling” or the “Reverse Warburg Effect”. Here, cancer cells induce mitophagy and make the associated glycolytic fibroblasts generate excessive lactate and ketones, which are taken up by the cancer cells from the extracellular space and used in mitochondrial energy production [88,89,90]. When cancer cells have to compete for nutrients with tumor-infiltrating immune cells, cell-intrinsic programs drive nutrient acquisition, where cancer cells depend on glutamine metabolism, leaving glucose for the immune cells [91].

Apart from ATP production, another important role of the TCA cycle for which functional ETC is needed is to provide oxaloacetate for the production of aspartate via aspartate aminotransferase (AAT). Aspartate is then exported to the cytosol for nucleotide biosynthesis for proliferation, as well as to provide NAD^+^. Cells rely on glutamine for this as, in glutamine-deprived conditions, the cytosolic aspartate availability determines the cell survival [92,93,94,95]. Cancer cells often downregulate the aspartate-consuming enzyme argininosuccinate synthase (ASS1) in the urea cycle to increase the aspartate availability for nucleotide biosynthesis [96]. Another role of respiration-derived aspartate is the production of asparagine for maintenance of the mTOR complex I (mTORC1) for tumorigenesis. The inhibition of respiration in a combination of treatments that limit the environmental asparagine availability, like asparaginase or dietary asparagine restriction, has shown therapeutic potential in mouse models of lung, breast and pancreatic cancers [97]. Moreover, the ability of breast cancer cells to synthesize asparagine when the bioavailability is compromised is strongly correlated with its metastatic potential [98]. Another way ETC is interlinked with nucleotide biosynthesis is through the role of complex III in the oxidation of ubiquinol to ubiquinone, which is an electron acceptor for dihydroorotate dehydrogenase (DHODH), an enzyme essential for de novo pyrimidine synthesis [99].

Due to aggressive cell proliferation, the tumor size extends beyond the natural diffusion limits of oxygen, and the core of the tumor becomes hypoxic. Even though tumors acquire blood vessels by inducing angiogenesis, most solid tumors retain hypoxic domains throughout disease progression, which, in effect, selects for aggressive malignant cells [100]. The exposure of hypoxia increases mitochondrial ROS generation by ETC, which is essential for hypoxia signaling. Upon oxygen sensing, mitochondrial ROS is generated via the Rieske iron–sulfur protein (RISP) of complex III. This process is independent of OXPHOS, as lacking complex III subunit cytochrome b impairs OXPHOS, with no impact on oxygen sensing [101,102]. Hypoxia increases the release of a superoxide from mitochondrial complex III, which is converted into H_2_O_2_ by SOD1 and inhibits prolyl hydroxylase (PHD), resulting in HIF-1α (hypoxia inducible factor-1α) stabilization [103]. In normoxia, PHD catalyzes the hydroxylation of two proline residues in the oxygen-dependent degradation (ODD) domain of HIF-1α, promoting its subsequent degradation by VHL (von Hippel-Lindau protein). Loss of the VHL gene associated with ~70% clear cell renal cell carcinoma contributes to tumorigenesis by the HIF-mediated induction of hypoxia response elements (HRE) [104,105]. The mechanism by which ROS inhibits PHD is yet to be defined. Several mechanisms have been proposed, one of which might be inducing a shift in the iron redox state from Fe^2+^ to Fe^3+^ (Fenton reaction), thereby limiting an essential cofactor of the PHDs [106]. Another way of inhibition is via the oxidative modification of PHD’s cysteine residue as, when the free intracellular cysteine level is high, PHD2 cysteine oxidation is prevented, keeping it active and the HIF-1α level low. When the intracellular cysteine level is depleted due to the glutamate-induced inhibition of xCT glutamate–cystine antiporter in TNBC (triple-negative breast cancer), HIF-1α is stabilized in a normoxic condition due to the oxidative self-inactivation of PHD [107,108].

The accumulation of mitochondrial metabolites, succinate and fumarate in SDH and FH-deficient cancer inhibits PHD by competing with its substrate α-ketoglutarate and causes the stabilization of HIF-1α in a normoxic condition, creating a pseudohypoxic condition [109]. These tumors are glycolytic, and for generating TCA cycle intermediates, they employ part of the oxidative TCA cycle, leading up to the deficiency and other metabolic pathways, which differ from one another. FH tumors utilize glutamine-dependent “reductive carboxylation”, and SDH tumors have robust pyruvate carboxylase activity to generate oxaloacetate for nucleotide synthesis [110,111]. Similar pseudohypoxic conditions may also prevail due to the accumulation of competitive antagonist 2-hydroxyglutarate (D-2HG) in IDH2 mutant cancers and in acidic pH, which enhances the production of L-2HG via the noncanonical activity of lactate dehydrogenase [112].

Once HIF-1 is activated, it promotes the transcriptional activation of a set of genes that carries out the adaptive process for survival under chronic hypoxia. Part of this process is survival in reduced oxygen tension, feedback and dampen mROS, metabolic reprogramming, angiogenesis, invasion and metastasis, critical for tumor progression [100,113]. HIF-1 induces the subunit switch of cytochrome C oxidase (COX) that optimizes the efficiency of respiration at a lower oxygen concentration [114]. Depending on the cell type, hypoxia brings about changes in the mitochondrial morphology, inhibits mitochondrial biogenesis by inhibiting Myc activity and induces mytophagy via mitochondrial outer-membrane protein FUNDC1, which, in combination, reduces mROS generation [115,116,117]. Hypoxia suppresses the cellular ATP consuming processes via triggering the activation of AMPK (AMP-activated protein kinase) through the ROS-mediated opening of calcium release-activated calcium (CRAC) channels, leading to an increase in cytosolic calcium that activates the AMPK via upstream kinase CaMKKβ (Ca^2+^/calmodulin-dependent protein kinase kinase β) [118].

Hypoxia signaling changes the metabolic landscape, resulting in diverting carbon entry to the mitochondria and reducing the TCA cycle flux to ultimately suppress the activity of the ETC (Figure 2). Pyruvate generated from glucose is secreted as lactate via the HIF-mediated induction of lactate dehydrogenase A (LDHA) and pyruvate dehydrogenase kinase 1 (PDK1), which phosphorylates and inactivates pyruvate dehydrogenase (PDH) [119,120]. Glutamine is metabolized via cytosolic reductive carboxylation [121,122] (Figure 2). The mechanism by which HIF1 induces reductive carboxylation is via SIAH2-targeted ubiquitination and proteolysis of the oxoglutarate dehydrogenase complex (OGDH2), leaving glutamine-derived α-ketoglutarate to be carboxylated by IDH1 [123]. HIF-1 also mediates a metabolic switch that inhibits fatty acid beta oxidation and promotes fatty acid synthesis, contributing to tumorigenesis [123,124]. An important consequence of a reduced TCA cycle flux is the reduction in aspartate biosynthesis due to the lack of the substrate oxaloacetate, which may impair tumor growth. Cancer cells import it via a cell-specific SLC1A3, an aspartate/glutamate transporter, or synthesize it from cytosolic oxaloacetate via GOT1 (glutamate-oxaloacetate transaminase 1) [94,125]. For the generation of oxaloacetate and lipids, they depend on glutamine-derived reductive carboxylation and may increase its uptake [126]. Glutamine dependency opens a therapeutic opportunity based on the systemic administration of glutaminase inhibitors [127]. Hypoxia stimulates acetyl-CoA synthetase 2 expression, which catalyzes the production of acetyl-CoA from acetate, making it a major carbon source in breast and prostate cancer cells for lipid synthesis [128]. Hypoxia may also suppress the activity of fatty acyl desaturase SCD, which may render the toxic accumulation of a saturated fatty acid precursor, and uptake of an unsaturated lipid becomes important for maintaining the homeostasis in hypoxia [129,130]. Hypoxia induces mitochondrial fission via upregulating the expression of dynamin-related protein 1 (Drp1). This contributes towards gaining aggressive metastatic characteristics [131,132].

To mitigate the increased ROS produced during hypoxia, glucose carbon is diverted towards serine and glycine biosynthesis in the cytosol via the increased expression of phosphoglycerate dehydrogenase (PHGDH). This renders an increased glutathione biosynthesis and NADPH production via one-carbon metabolism in the folate pathway [133,134] (Figure 2). Serine hydroxymethyl transferase 2 (SHMT2), the first enzyme in the mitochondrial one-carbon metabolism pathway, mediates serine degradation to produce NADPH in hypoxia in Myc-transformed cells. Its expression is correlated with an unfavorable patient prognosis. Moreover, there is a positive correlation between mitochondrial serine catabolism and cytosolic de novo serine biosynthesis, indicating the contribution of the cytosolic serine biosynthesis pathway in maintaining redox metabolism in the mitochondria [135] (Figure 2). Serine regulates carbon distribution in the anabolic vs. biosynthetic route through the allosteric control of pyruvate kinase M2 (PKM2), overexpressed in cancer cells. Serine deprivation reduces PKM2 activity, causing the accumulation of a precursor for serine biosynthesis: 3-phosphoglycerate [133]. The constitutive activation of PKM2 renders cancer cells dependent on exogenous serine [136]. Since serine biosynthesis requires NAD^+^, cancer cells expressing high levels of PHGDH are dependent on the NAD^+^ salvage pathway for their generation, creating a therapeutic potential for inhibitors of this pathway [137]. Serine-deprived cells are sensitive to conditions that decrease the cellular NAD^+^/NADH ratio [138].

Increased collagen matrix formation and ECM (extracellular matrix) stiffness are associated with tumor growth and progression in different cancers [139]. The role of HIFs in inducing the expression of both collagen and extracellular matrix remodeling enzymes that promote the aberrant collagen deposition and acquisition of the prometastatic local extracellular matrix has been well-defined [140,141]. Nearly 25% of the amino acids incorporated into collagen are proline, thus indicating the important role of proline metabolism in collagen synthesis and ECM remodeling. Oncogene *c-Myc* has been shown to increase proline biosynthesis for maintaining the redox balance [142,143]. The accumulation of hydroxyproline is also important for promoting hypoxia phenotype [144]. Mitochondrial NADPH is required for generating pyrroline-5-carboxylate, a bottleneck in proline biosynthesis, and this makes cells dependent on NADK2 for the generation of NADP^+^ in mitochondria [31]. ECM stiffening promotes kindlin-2 translocation into the mitochondria and interacts with PYCR1 (Pyrroline-5-Carboxylate Reductase 1) for proline biosynthesis [145]. In cancers associated with collagen-rich meshwork like pancreatic ductal adenocarcinoma, collagen serves as a reservoir for proline, which can be taken up in a nutrient-limited condition. One mechanism that emerges from this is mitochondrial-reducing equivalents that are consumed during proline biosynthesis and exported for collagen formation, which can then be imported and catabolized to support TCA cycle metabolism and provide reducing equivalents for cellular respiration [146].

SIRT3, a master regulator of mitochondrial metabolism, suppresses HIF-1α and tumor growth by inhibiting mitochondrial ROS production from complex III [147]. The tumor suppressive role of SIRT3 is mainly mediated by reversing the Warburg-like metabolic reprogramming [148].

## 5. Metastatic Dissemination

The role of mitochondria in metastasis first came to attention from a swap experiment where the mtDNA of highly metastatic tumor cells conferred enhanced aggressiveness and metastatic potential owing to the production of mitochondrial ROS [149]. This is pronounced in cells deficient in mtDNA incapable of generating mitochondrial ROS that fail to grow in an anchorage-independent manner [55]. Enhanced superoxide production from mitochondrial ETC either by overload or partial inhibition promotes various aspects of metastasis via protein tyrosine kinases Src and Pyk2 as downstream effectors [150].

While most studies focus on the overall success of metastasis by observing distant colonization, the dynamic metabolic adaptation during the different phases of the metastatic cascade remains to be understood [151]. The small subpopulation of cells that can successfully proceed through the stages of the metastatic cascade is difficult to trace and isolate, making analyzing their metabolisms challenging. Technical advancements in experimental models representing an endogenous microenvironment and model organisms could help gain insight into the metabolic alterations that must occur to proceed with each step [152,153]. Metastatic cancer cells arising from different primary tumor sites may have site-specific metabolic landscapes, which start changing once they are detached from the primary site and invade depending on the constraints of the originating tissue. They may tend to converge towards a specific metabolic signature when they arrive in the circulation [9]. Such a type of converging metabolic adaptation is an attractive avenue for potential therapeutic interventions.

The delamination of cells from the primary tumor, representing the first step of the metastatic cascade, occurs through a process similar to epithelial-to-mesenchymal transition (EMT) [154,155]. It is a reversible phenomenon that allows epithelial cells to become motile and invade adjacent tissues to enter the circulation [156]. Mitochondrial ROS plays an important signaling role in inducing EMT [157,158]. Matrix metalloproteinase 3 (MMP3) is upregulated in many cancers and responsible for the induction of EMT and exerts its action by expression of an alternatively spliced form of Rac1, which causes mitochondrial ROS accumulation [159]. Mitochondrial metabolites, like 2HG, succinate and fumarate, accumulated in different cancers associated with mutations in enzymes producing them, are involved in the transcription of a set of genes that carries out EMT by inhibiting histone and DNA demethylases [160]. Similar phenomena happen during the accumulation of acetyl-CoA due to suppressed mitochondrial oxidation, where acetyl-CoA induces histone acetylation, leading to open chromatin and the expression of genes associated with EMT [161]. The proline hydroxylation of collagen during ECM remodeling for the progression of EMT requires vitamin C, which limits its availability for DNA and histone demethylases, increasing the overall DNA and histone methylation and causing a transition in the cellular state in EMT [162,163].

For efficiently metastasizing, the cancer cell must develop a resistance to anoikis, defined as the induction of apoptosis owing to detachment from the extracellular matrix (ECM), and amorphosis, cell death stimulated by loss of the cytoskeletal structure [164,165,166,167]. Cancer cells decrease glucose oxidation following detachment for anoikis resistance, and when glucose oxidation is stimulated through inducing PDH in suspension, it makes them susceptible to anoikis due to increased mitochondrial ROS production. Therefore, suppressing mitochondrial oxidative reactions that, in effect, limit mitochondrial ROS generation is one mechanism for cells to become resistant to anoikis [168,169]. Fatty acid oxidation mediated by CPT1A is important for eliminating ROS, rendering colorectal cancer cells resistant to anoikis [170].

To check the accumulation of mitochondrial ROS associated with anchorage-independent growth, glucose and glutamine metabolism is modulated with two main goals, decreasing mitochondrial oxidation for minimizing the production of ROS and increasing mitochondrial NADPH pool for mitigating the ROS (Figure 3). An important phenomenon observed here is the contribution of a cytosolic NADPH pool in reductive carboxylation for supporting NADPH production in the mitochondria for mitigating mitochondrial ROS [43]. NADK2 plays important role in this stage to produce NADPH in mitochondria [31]. When cells start to get detached from the extracellular matrix, there is a loss of glucose transport. Shunting a considerable amount of glucose through PPP for generating NADPH the in cytosol becomes critical to support ROS detoxification and mitochondrial NADPH production [171]. The induction of PPP for NADPH production is mediated through increased Nrf2 expression, which is correlated with cancer metastasis [172,173]. Matrix deprivation leads to a spike in the intracellular calcium, which triggers the CaMKKβ-mediated activation of AMPK signaling, leading to ATP conservation and anoikis resistance [174,175]. AMPK also inhibits acetyl-CoA carboxylases ACC1 and ACC2, inhibiting fatty acid synthesis and sparing NADPH for maintaining the redox homeostasis [176]. Upon cell detachment, there is increased dependency on SOD2 for the detoxification of mitochondrial superoxide. The expression of SOD2 correlates with histologic tumor grades, further supporting its role during metastasis [177]. Even though Sirt3 shows a tumor-suppressive role during growth, it has a critical role during matrix detachment and colonization. In ovarian cancer, Sirt3 expression and activity transiently increases in detached cells for mitigating mitochondrial superoxide surges by regulating SOD2 [178]. Various oncogene-initiated signal transductions also modulate the mitochondrial metabolism for adaptation, like Ras activation induces a distinct PI(3)K effector, serum and glucocorticoid-regulated kinase-1 (SGK-1) for regulating the mitochondrial function to overcome the ATP deficiency induced by ECM detachment [179,180].

In a genome-scale metabolic model of the NCI-60 cell lines, the glycolytic-to-oxidative ATP flux ratio was found to be highly positively correlated with cell migration, indicating that ATP from glycolysis, rather than oxidative phosphorylation, supports cancer cell migration and more aggressive metastatic tumor development. Silencing genes that target this ratio are good candidates for developing a therapy that targets more aggressive metastatic cells without the side effects associated with current antiproliferative treatments [181]. During cell migration and isogenic cell lines carrying varying degrees of mtDNA mutation impacting oxidative phosphorylation, where an enhanced reliance on glycolytic ATP synthesis is observed, reductive carboxylation has been proven to support the glycolytic flux via recycling cytosolic NADH. Malate dehydrogenase 1 (MDH1) is a novel link between reductive carboxylation and glycolysis and may interact with GAPDH, which helps enhance glycolytic NADH recycling (Figure 3) [182]. Mitochondrial fission mediated by Drp1 is required for lamellipodia formation during cell migration [131,183].

Discrepancies are observed in the bioenergetic profile of the mitochondria during metastasis in different cancers. Invasive breast cancer cells employ transcriptional coactivator PGC-1α (peroxisome proliferator-activated receptor gamma, coactivator 1 alpha) to enhance oxidative phosphorylation and mitochondrial biogenesis, in effect relying on mitochondrial respiration during metastasis [184]. A completely opposite phenomenon is observed in prostate cancer, where the downregulation of PGC-1α is associated with cancer progression and metastasis [185]. Alteration of the fuel preference is also observed in metastasis, as metastatic TNBC cells rely on fatty acid beta oxidation for an ATP requirement [186]. The expression of fatty acid receptor CD36 has been associated with metastasis initiation in numerous types of carcinoma, and they rely on beta oxidation from palmitic acid or a high fat diet for successful metastasis [187].

During metastatic spread, mitochondrial ROS accumulation presents an additional challenge in combination with the stressful environment of the circulation, where blood represents elevated oxidative stress compared to lymph [188]. Clustering of the detached cells accelerates the metastatic spread and survival in the circulation. Cell clustering limits ROS by driving hypoxia and HIF-1α-mediated mitophagy, thus removing damaged ROS-producing mitochondria. The resultant decrease in the mitochondrial capacity results in a dependence on glycolysis that is supported by the reductive carboxylation of glutamine to malate [189]. Studies on metastatic melanoma cells shed light on how metastatic cells maintain the redox balance in the circulation. In the circulation, these cells depend on the folate pathway to generate NADPH, and it opens an option for therapeutic intervention with methotrexate or via genetic alteration [190]. Similar studies with a mouse model of malignant melanoma showed the beneficial effect of antioxidants to promote metastasis [191]. Efficient metastatic melanoma cells depend on MCT1-dependent lactate uptake, which serves as a fuel for mitochondrial oxidative metabolism [192].

## 6. Distant Colonization

Once the circulating tumor cells (CTCs) exit the circulation at their metastatic site, they again have to alter their metabolism to adapt to the new environment of the host organ. Out of the millions of cancer cells shed in the circulation by primary tumors, only a very small portion could successfully withstand the harsh environment of the circulation and have the necessary adaptive capabilities to colonize and form metastatic lesions in distant organs [9]. As anoikis also prevents cells from attaching to foreign or inappropriate matrixes, developing a resistance or inhibition to anoikis is important for colonization at distant organs [193].

Different tumor-secreted factors and tumor-shed extracellular vesicles actively modify the organs of future metastasis to create a premetastatic niche amenable for the growth of metastatic cancer cells [194,195,196]. Once the metastatic cancer cells reach their targeted metastatic organ, they must actively breach the vasculature in a process called extravasation to leave the circulation and colonize [197]. The increased accumulation of cellular ROS plays a major role in extravasation by inducing signaling pathways like ERK1/2 [198]. Upon anchorage to the organ of metastasis, incoming metastatic cells require undergoing reversing the EMT process required for invasion to MET (mesenchymal-to-epithelial transition) for colonization [199,200,201].

In the circulation, metastatic cells are exposed to oxygen limited conditions, which hampers their energy production. For colonizing in the targeted metastatic site, cancer cells actively modify the metabolism of the host organ to utilize them for the generation of energy. Cancer cells secrete microRNA and actively modify the energy metabolism of recipient premetastatic niche cells by downregulating pyruvate kinase, in effect decreasing the glucose uptake and promoting glucose availability for metastatic cells [202]. This process is further aided by hypoxia signaling [203]. This phenomenon is observed when colorectal cancer cells metastasize to the liver. Upon metastatic dissemination into the liver, colorectal cancer cells experience hepatic hypoxia and secrete creatine kinase, brain-type (CKB) in the extracellular matrix of the liver, which catalyzes the generation of high-energy phospho-creatine and is taken up by the metastatic cell for the production of ATP [204].

The microenvironment of the distant organ also plays role in colonization of the metastatic cells [205]. A hypoxic microenvironment often selects disseminated metastatic cells that have the ability to withstand hypoxic stress and metabolically adapt to colonize in distant organs [203]. How organ-specific metastatic traits arise in primary tumor cells remains an area of active research, and many factors are emerging that play a critical role [206]. Mitochondrial redox metabolism plays a central role in determining the adaptability of tumor cells according to the metabolic constraints of the metastatic site. Some primary tumors only metastasize to specific organs, like prostate cancer to the bone and pancreatic cancer to the liver, while others are capable of metastasizing in different organs, like breast and lung cancers [207]. The heterogeneity of the mitochondrial metabolism within breast cancer cells determines their site of metastasis, like liver metastatic cells display increased HIF-1α activity and lactate production with decreased mitochondrial oxidative metabolism compared to bone and lung metastatic cells [208].

The rewiring of the mitochondrial metabolism in metastatic cells and preference for fuel also depend on the nutrient availability of the distant organ where metastasis occurs, like the availability of pyruvate in the lungs activates pyruvate carboxylase in the metastatic breast cancer fueling TCA cycle [209]. They also rely on pyruvate metabolized by part of the TCA cycle to produce α-ketoglutarate for collagen hydroxylation to actively modify the ECM for creating a lung metastatic niche. Worth mentioning is that importing pyruvate instead of lactate would be preferable in cellular NAD^+^-restricted conditions, indicating the redox status of the incoming metastatic cells [210]. Another important pathway for fueling the metastatic growth of breast cancer in the lungs is proline catabolism via proline dehydrogenase generating FADH2 [211].

Organ-specific metabolic constrains mainly depend on the tissue architecture, microenvironment, nutrient availability, redox status and activity of the organ [212]. A low glucose concentration in the brain interstitial space requires incoming metastatic cells relying on the oxidation of glutamine and branched-chain amino acids [213]. Cancer cells that metastasize in the brain from a wide variety of primary tumors, like breast cancer, non-small-cell lung cancer (NSCLC), clear-cell renal cell carcinoma, melanoma and endometrial cancer, also use acetate as a bioenergetic substrate to fuel growth, which is a brain-specific adaptation [214]. Breast cancer cells metastasizing into the brain may also show a GABAergic phenotype, which renders them able to catabolize GABA into succinate with the resultant formation of NADH [215].

As the lungs are the primary organ for respiration in humans, they are exposed to high levels of oxygen, as well as toxic compounds, which contribute to increased oxidative stress, presenting a challenge for metastatic cells to colonize [216]. To withstand and survive in this oxidative stress condition, mitochondrial complex I is modulated and antioxidant programs activated in breast cancer cells, forming micrometastasis in the lungs [217,218]. PGC-1α-mediated transcriptional regulation associated with increased mitochondrial biogenesis is required for breast cancer cells to metastasize in the lungs, possibly through increasing the overall efficiency of the ETC machinery in the oxidative conditions of the lungs and reducing ROS generation [184,219].

The liver is a key organ for maintaining the energy balance of the body by regulating the blood glucose level through tightly regulating the glucose consumption and production. The liver is also divided in metabolic zonation, which corresponds to varying oxygen gradients. The liver microenvironment is naturally more conducive to cells that display a high glycolytic profile and are adapted to a low-oxygen state, as seen in studies with primary hepatocellular carcinoma showing a preference for engaging in glycolytic metabolism [220]. Breast cancer cells that metastasize in the liver upregulate PDK-1, causing a decrease in the TCA cycle flux and overall suppress mitochondrial oxidative metabolism and increase hypoxia signaling [208].

During bone metastasis, invading cells mostly induce osteolytic mechanisms as destruction of the niche tissue renders more nutritional sources available. Regulatory pathways that modulate the bone metabolism, such as bone growth and resorption, are, therefore, likely to be key in determining whether cancer cells can colonize to the bone and establish metastatic lesions. Osteoclasts responsible for bone resorption require serine, making it critical for osteolytic bone metastasis. Bone metastatic breast cancer cells express high levels of serine biosynthesis genes for efficient metastasis, and since serine biosynthesis requires a supply of oxidized NAD^+^, this may indicate that the cellular redox status determines the efficiency of bone metastasis [138,221]. Another feature of highly glycolytic bone metastatic breast cancer cells is that they secrete large amounts of lactate, which is taken up by osteoclasts via the MCT1 transporter for fueling its oxidative metabolism, rendering them highly active [222].

## 7. Concluding Remarks

Emerging studies are pointing to the fact that the metabolic needs and vulnerabilities of cancers change throughout different stages of tumor development, from tumor initiation and growth to metastasis and colonization. The central role of mitochondrial redox metabolism governing the reprogramming and adaptability of metabolic processes during cancer progression makes it a prime target for developing therapeutic strategies (Table 1).

There has been a lot of efforts in recent years to target different aspects of mitochondrial metabolism for cancer treatments (Table 2). These efforts mainly fall under three avenues: targeting bioenergetics, biosynthetic and redox metabolism of the mitochondria [223]. Targeting mitochondrial ATP production could be effective for cancers that rely on mitochondrial oxidative phosphorylation or in poorly perfused tumors. Efforts to decrease mitochondrial ROS production have shown limited potential due to diminished localized effects. Contrary to this, recent studies have demonstrated that targeting antioxidant machineries has shown greater potential as an anticancer therapy. This approach relies on the cytotoxic effect of enhanced ROS accumulation when the antioxidant capacity is compromised in cancer cells [224].

We still need a plethora of works to decode the reprogramming of metabolism that happens during each stage, which will elucidate vulnerabilities for stage-specific metabolism-targeted therapies. Tumor metabolisms are heterogeneous between different types of cancers. Distinct region-specific metabolic reprogramming within a tumor adds to the complexity and presents an additional challenge for finding a therapeutic target. Once cancer cells disseminate from primary tumor sites, they converge towards a specific metabolic characteristic for survival in the circulation and colonize in a distant organ. Targeting this critical stage might be a promising therapeutic strategy for controlling metastasis, which is the primary cause of cancer-related mortality.

## Figures and Tables

**Figure 1 antioxidants-10-01838-f001:**
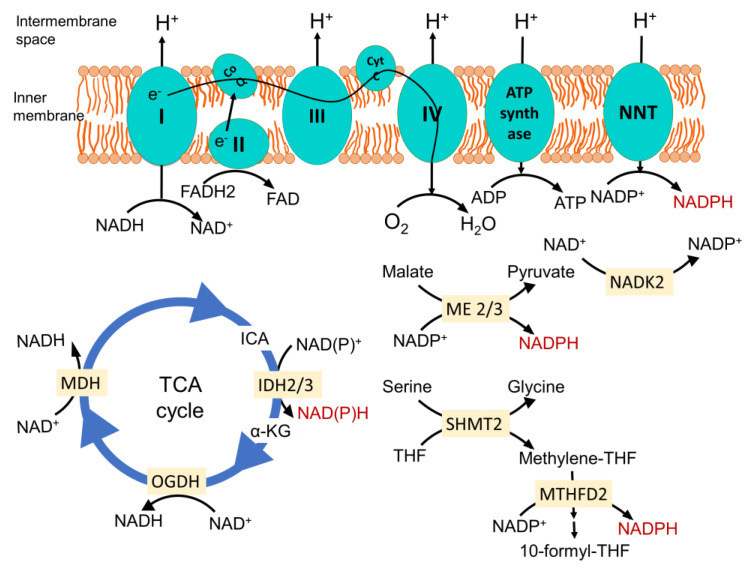
Mitochondrial major NADPH-producing pathways. Nicotinamide nucleotide transhydrogenase (NNT) located in the inner membrane produces NAPDH from NADP^+^. It is energized by the inner membrane electrochemical gradient, produced by proton translocation via the electron transport chain (ETC) during the sequential flow of electrons from reducing equivalent NADH and FADH_2_ to O_2_ to form H_2_O. This enzyme plays a pivotal role in dedicating the mitochondrial NADH pool towards maintenance of the optimum NADPH level. Isocitrate dehydrogenase 2 (IDH2) in the TCA cycle catalyzes the oxidative decarboxylation of isocitrate to α-ketoglutarate, and malic enzymes 2 and 3 (ME2 and ME3) catalyze the oxidative decarboxylation of malate to pyruvate while generating NADPH from NADP^+^ in the process. In the one carbon metabolism once serine is catabolized by serine hydroxymethyl transferase 2 (SHMT2), it generates methylene-THF. The NAD kinase (NADK2) phosphorylates NAD^+^ to NADP^+^, feeding other NADPH-generating reactions. Methylene tetrahydrofolate dehydrogenase 2 oxidizes it to methenyl tetrahydrofolate and, finally, to 10-formyl-THF, which is coupled to the reduction of NADP^+^ to NADPH. CoQ, coenzyme Q; Cyt C, cytochrome C; ICA, isocitrate; α-KG, α-ketoglutarate; OGDH, oxoglutarate dehydrogenase; MDH, malate dehydrogenase; THF, tetrahydrofolate; I, II, III and IV represents complexes I, II, III and IV in the mitochondrial electron transport chain.

**Figure 2 antioxidants-10-01838-f002:**
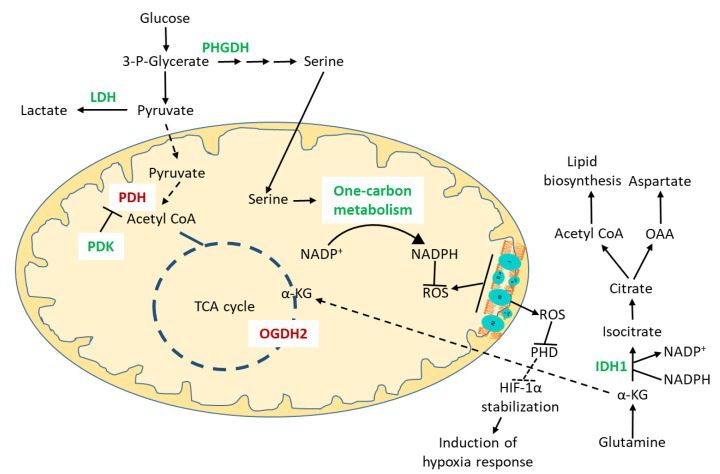
Metabolic reprogramming supporting the redox balance in mitochondria during hypoxia. ROS (reactive oxygen species) generated by the electron transport chain in the mitochondrial matrix during hypoxia. The Rieske iron–sulfur protein (RISP) of complex III generates ROS in the cytosol, which inhibits prolyl hydroxylase (PHD), resulting in HIF-1α (hypoxia inducible factor-1α) stabilization followed by the induction of various hypoxia responses. Seine catabolized in the mitochondria in one-carbon metabolism to generate NADPH to mitigate ROS in the mitochondria in Myc transformed cancer cells. Cytosolic serine biosynthesis is upregulated via the upregulation of phosphoglycerate dehydrogenase (PHGDH) to support mitochondrial serine catabolism. Glucose carbon is diverted to lactate production, decreasing entry into the mitochondria and reducing the TCA cycle flux. The induction of a hypoxia response causes the inhibition of oxoglutarate dehydrogenase (OGDH2), rendering α-ketoglutarate (α-KG) to be carboxylated by isocitrate dehydrogenase 1 (IDH1) in the cytosol, contributing to reductive carboxylation supporting the lipid and nucleotide biosynthesis (via aspartate) in hypoxia. Dashed lines represent downregulation. Green and red represent up and downregulated enzymes and pathways as part of the hypoxia response. OAA, oxaloacetate; 3-P-Glycerate, 3-phosphoglycerate.

**Figure 3 antioxidants-10-01838-f003:**
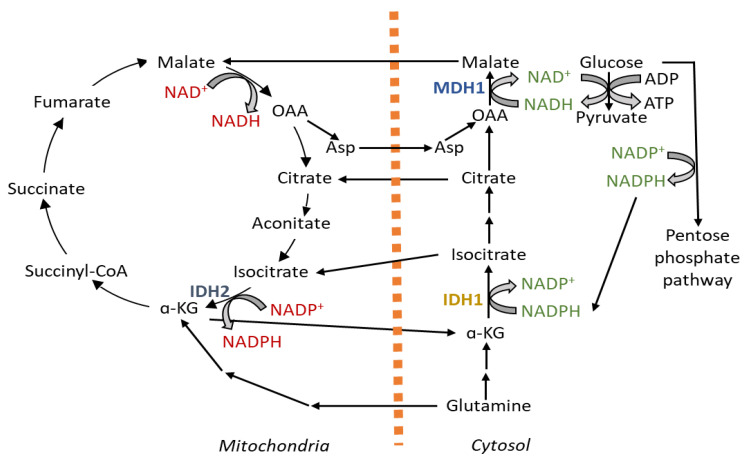
Cytosolic reducing equivalents supporting the mitochondrial redox balance during metastatic dissemination. Glucose and glutamine metabolisms are attenuated during metastatic dissemination to mitigate the accumulation of mitochondrial reactive oxygen species (ROS). Glutamine goes through reductive carboxylation, where isocitrate dehydrogenase 1 (IDH1) consumes NADPH from the cytosol, and citrate generated in this process enters into the mitochondria to support NAPDH production via isocitrate dehydrogenase 2 (IDH2) and fortify the ROS defense. Glucose is metabolized through the pentose phosphate pathway to provide cytosolic NAPDH needed by IDH1. Oxaloacetate (OAA) produced via reductive carboxylation is metabolized by malate dehydrogenase 1 (MDH1) in the cytosol to produce NAD^+^ to support the glycolytic flux. α-KG, α-ketoglutarate; Asp, aspartate.

**Table 1 antioxidants-10-01838-t001:** Key aspects of mitochondrial redox metabolism during different stages of cancer progression.

Mechanism/Features	Effects
**Malignant transformation**	
Mitochondrial DNA mutations	Impair oxidative phosphorylation [48,49,51]
	Increase in ROS production [48,49,50]
	ROS-induced inactivation of PTEN and activation of the PI3K pathway [53,54]
	Inhibition of ROS-induced apoptosis [66]
Defect in MnSOD	Increase in genomic instability mediated by ROS [60,61]
Oncogenic Kras activation	Increase in mitochondrial ROS generation and metabolic switch from oxidative phosphorylation to glycolysis [55,56]
**Tumor development**	
Impaired oxidative phosphorylation	Increase in glycolytic flux for ATP synthesis [70,79]
Tumor specific bioenergetic pathways	Govern the type of fuel consumption by specific tumors [81,84]
Hypoxia	Mitochondrial ROS-mediated HIF-1α stabilization [101,102,103]
	ROS-induced metabolic reprogramming enhance the reliance on glycolysis and reductive carboxylation [119,120,121,122,123]
	NADPH production via serine degradation in mitochondria to mitigate ROS [135]
	Increase in collagen matrix formation and extracellular matrix (ECM) remodeling via proline metabolism [142,143,144,145,146]
**Metastatic dissemination**	
MMP3 upregulation	Mitochondrial ROS-mediated induction of epithelial-to-mesenchymal transition (EMT) [157,158,159]
Inhibition of mitochondrial oxidative reactions	Decrease in mitochondrial ROS production conferring resistance to anoikis [168,169]
Stress associated with detachment from ECM	Cytosolic NADPH is consumed in reductive carboxylation to support NADPH production in the mitochondria to fortify the ROS defense [43,171,172,173]
Clustering of the detached cells	Decrease mitochondrial ROS generation via hypoxia signaling-mediated mitophagy and glycolysis induction [189]
**Distant colonization**	
Pyruvate metabolism by part of TCA cycle	Produce α-ketoglutarate for collagen hydroxylation to modify ECM for creating metastatic niche [209,210]
Heterogeneity of mitochondrial metabolism	Dictate adaptability when colonizing in specific organs, like breast cancer cells metastasizing in the liver, have decreased mitochondrial oxidative metabolism compared to bone and lung metastatic cells [208]

**Table 2 antioxidants-10-01838-t002:** Notable anticancer agents targeting mitochondrial metabolism.

Anticancer Agents	Target Site	Effect on Mitochondrial Metabolism
Metformin/Phenformin	Complex I	Decrease ATP production [225]
VLX600	ETC	Decrease ATP production [226]
C-968	Glutaminase	Suppress contribution of glutamine to the TCA cycle [227]
Chloroquine	Autophagy	Suppress contribution of autophagy to the TCA cycle [228]
ATN-224	SOD1	Increase accumulation of ROS causing cell death [224]
CPI-613	TCA cycle	Induce mitochondrial ROS production causing cell death [229]
Leflunomide	DHODH	Inhibit ETC-dependent pyrimidine biosynthesis [230]

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
