# Peer review of "Mitochondrial Redox Metabolism: The Epicenter of Metabolism during Cancer Progression"

_antioxidants, 2021, doi:10.3390/antiox10111838_

Round 1

Reviewer 1 Report

In this review Dr. Choudhury addresses the mitochondrial redox metabolism during cancer progression. The metabolic changes that cancer cells adopt to face the different stages of cancer progression are nowadays actively studied to highlight possible stage-specific metabolic vulnerabilities that could become future therapeutic targets to tackle cancer. Despite the interesting topic, the manuscript is not suitable for publication because the text is poorly written and poorly organized and needs to be reworked.

General comments:

  • an extensive editing of English language, style and punctuation is required to improve the readability of the manuscript to be suitable for publication.
  • Please provide a table/box summarizing the main “redox” characteristics of the different stages of tumor progression. This would be highly beneficial for Antioxidants Journal readership and would maximize the impact of this review article.
  • Many sentences are too cryptic and need clarification. For example: line 74, explain what “retrograde signaling” means. Lines 75-78, explain better. Please, check the entire manuscript carefully.

Introduction:

The introduction is repetitive. A short introduction works best for reviews. To improve the introduction, the author should try to embed the information contained in lines 41-47 into the first part, as these lines don’t add much. Same for lines 51-54.

Paragraphs 3-6:

The text is poorly structured. Adding subparagraphs may improve the readability. For example, paragraph 2 may be split into two sections addressing pathological/aberrant versus physiological/regulated production of ROS. Also, please remove (or reduce) information that is out of the scope of the review (i.e. REDOX metabolism). For example, lines 221-287, lines 336-357 and lines 497-519.

Reviewer 2 Report

This is a comprehensive overview of mitochondrial redox metabolism in the context of cancer disease. The author provides a good overview of published literature and key factors and molecules involved in the mitochondrial function (glycolysis as well). One possible area of improvement will be to add a short discussion (a few sentences and references) on the topic of mitochondrial plasticity (since mitochondrial fission and fusion seems to play critical roles in maintaining functional mitochondria when cells experience metabolic or environmental stresses).

Reviewer 3 Report

The authors report a review manuscript untitled "Mitochondrial Redox Metabolism: The Epicenter of Metabolism during Cancer Progression".

Although there are similar reviews on the topic (e.g., https://pubmed.ncbi.nlm.nih.gov/29219147/), this review provides a great basis for the role of Mitochondria in cancer.

Mitochondria constitute promising targets for the development of novel anticancer agents. 

I am wondering if the author could include the know anticancer compounds targeting Mitochondria.

What is the known mechanism of action of these compounds targeting Mitochondria? Are they ROS suppressors?

A table will be highly informative for the journal audience.

references need to be updated with recent work on this topic:

https://pubmed.ncbi.nlm.nih.gov/31500275/

https://pubmed.ncbi.nlm.nih.gov/31500275/

Once the Author addresses these concerns, the review could be published.

Reviewer 4 Report

The authors present and interesting review of mitochondrial redox metabolism in cancer.  The paper is in need of proofreading and editing.  There are points that would benefit from clarification.  These are pointed out below.  Many of these issues would be picked up by standard editing software such as that included in Word.  Later in the manuscript the editing seems to be better. The authors should address the concerns listed below:

Line 17 – what “certain stage”  this sentence is a bit vague anda bit long  Can you elaborate and clarify?  Maybe make more than once sentence.

Line 33 – “P53-induced” instead of “P53 induced”

Line 34 – “stress-increasing” instead of “stress- increasing”

Line 36 – If ROS is associated with transformation there should not be a comma after ROS.

Line 37 – only keep the “,” after progresses the rest should be omitted.

Line 41 – Should be “The role”

Line 42 – “It is” instead of “It’s”

Line 44 – “on” instead of “one”.  Better to re-write “stages.  However, cancer cells need to develop….”

Line 56 – “namely” instead of “namely-“

Line 54-61 – sentence not clear – maybe break up into more than one sentence.

Line 66 – sentence not clear and should be reworded

Line 69 – “electrons” instead of “electron”

Line 70 – use either “increase membrane potential” or ‘’increased membrane hyperpolarization” instead of “maximum membrane polarization”

Line 71 – “and it lets” should be replaced with “causing”  and what are the electron carriers getting reduced?  Are you referring to the coenzyme Q being reduced to ubiquinol (QH2)?

Line 75 – “The Fenton” instead of “Fenton”

Line 114 = Use same reference in Figure legend and text (Fig 1 or Figure 1).

Super Oxide  is thought to be produced at complex I and III.  Complex I is into the mitochondria where it is converted to H202 which can cross the mitochondrial membrane through aquaporin water channels.

Line 155 – “pathways support” instead of “pathways supports”

Line 222 – “various aspects” instead of “various aspect”

Paragraph starting Line 272 – What is the role of the cytosolic and mitochondrial AAT (aspartate-amiontransferase?  How is aspartate replenished in the mitochondria?

Line 337 – Does the reduction of TCA cycle, reduce ROS production?  Is that important?

Line 347 – the pathway of aspartate biosynthesis is not clear.

Page 8-9 - Please make reference to Figure 2 when describing the roles of signaling by HIF-1 and PHDGH.

The Concluding remark are a bit short.  The authors might want to elaborate on status of the filed and challenges remaining.  Maybe a few words on the implications of the information in the review on clinical practice, drug discovery, etc.

Round 2

Reviewer 1 Report

The author has made some changes in response to my previous comments and improved her manuscript.

One point remains to be addressed

1) A table summarizing "mitochondrial redox metabolism and alteration during different stages of cancer progression" would be highly beneficial for Antioxidants Journal readership and would maximize the impact of this review article. I understand that preparing such a table/summary is demanding as the topic is wide, heterogenous and intricate. I suggest to include features that are typical of each stage of cancer progression and include appropriate references. For example:

Malignant transformation (Stage). Mitochondrial DNA mutations (Mechanism I). Increase in ROS production, inactivation of PTEN and activation of PI3K pathway (Effect I). Inhibition of ROS-induced apoptosis (Effect II) (ref 66). Increase in genomic instability (Effect III) (ref 54-57). And so on. 

Also, please check typos throughout the text.
